# Meanness trumps language: Lack of foreign language effect in early bilinguals' moral choices

**Albert Flexas***◉, **Raúl López-Penadés**◉◉, **Eva Aguilar-Mediavilla**‡, **Daniel Adrover-Roig**

Institute of Research and Innovation in Education (IRIE) Department of Applied Pedagogy and Educational Psychology, University of the Balearic Islands(Spain), Palma, Spain

◉ These authors contributed equally to this work.
‡ These authors also contributed equally to this work.
* albert.flexas@uib.eu

**Data Availability Statement:** All relevant data for this study are publicly available from the OSF repository (https://www.doi.org/10.17605/OSF.IO/NZRP3).

## Abstract

Moral decision-making is influenced by various factors, including personality and language. In this cross-sectional study, we investigated the Foreign-Language effect (FLe) in early, highly proficient, Catalan-Spanish bilinguals and examined the role of several personality dimensions in their responses to moral dilemmas. We obtained a multilevel data structure with 766 valid trials from 52 Catalan-dominant undergraduate students who read and responded anonymously to a computerized task with 16 standardized moral dilemmas, half in Catalan and half in Spanish. Results of a multilevel multivariate logistic regression analysis showed that consistent with previous research, participants gave more utilitarian responses to impersonal than personal dilemmas. The language of the dilemma had no effect on the response (dichotomous: utilitarian vs. deontological), decision time, or affective ratings, contradicting the hypothesis of shallower emotional processing of the information in the second language. Interestingly, cruelty features of psychopathy were significantly associated with an enhanced proportion of utilitarian decisions irrespective of the language or the nature of the dilemmas. Furthermore, cruelty features interacted with participants' assessment of dilemma aspects like vividness and verisimilitude. Overall, our findings suggest that early bilinguals immersed in a dual-language context using close Romance languages do not show the FLe and that personality traits like cruelty can modulate moral decisions regardless of language or dilemma type.

## Introduction

Pandemic times have raised evidence of the importance of moral decisions in our daily lives. Some of our actions can save or, perhaps, condemn others' lives. Factors like our personality and emotional implication in possible harm seem to be crucial to our moral decisions (see [1] for a systematic review and meta-analysis). In multilingual contexts, even language could make a difference in our moral decisions, maybe because language is unavoidably related to emotion [2–5] It has been said that bilinguals are more emotional about a moral dilemma

**Funding:** This study was financially supported by the Agencia Estatal de Investigación del Ministerio de Ciencia e Innovación (State Research Agency of the Ministry of Science and Innovation -MCIN/AEI- Spanish Government) and the European Regional Development Fund (ERDF) in the form of grants (EDU2017-85909-P and PID2021-123770OB-I00 funded by MCIN/AEI/ 10.13039/501100011033 and, by "ERDF A way of making Europe") received by all authors. No additional external funding was received for this study. The funders had no role in study design, data collection and analysis, decision to publish, or preparation of the manuscript.

**Competing interests:** The authors have declared that no competing interests exist.

when it is presented in their first language than their second language (see [6] for a review). However, regarding this so-called Foreign-Language effect (FLe), there is certain controversy about their presence in early, highly proficient, bilinguals (e.g., [2,7–9]), even in late proficient bilinguals (e.g., [7,10]). The present study aims to clarify the presence of the FLe in early high proficient bilinguals and investigate the role of other potentially relevant factors, such as certain personality traits when participants answer moral dilemmas.

Moral dilemmas are hypothetical short stories used broadly for research on moral reasoning [11]. This kind of stimuli describes a situation in which participants are confronted with conflicting moral reasons, such as the contrast between adhering to a deontological principle (e.g., not to kill) versus maximizing overall welfare through utilitarian means (e.g., kill one to save many). In the last years, research on moral cognition has provided valuable data to understand the processes underlying these competing reasons and how we adjudicate between them. Following Kahneman [12], Greene et al. [13–15] proposed that the decision to a moral dilemma relies on a dual process model (DPM), according to which an automatic strong affective response directed to avoid harm contrasts with a controlled reasoning process tending to suppress the emotion to solve the problem with the lesser harm. Thus, certain situational factors, such as the personal force variable, affect the response. Personal dilemmas in which a person is involved directly in harming others, could be high-conflict dilemmas, emotionally arousing, and move the person to respond deontologically (do not harm). Conversely, less emotional situations, like impersonal dilemmas in which there is some mechanism or distance between the protagonist and the harm (diminishing internal conflict), let the person think rationally and respond utilitarianly (harm to prevent bigger harm). In this vein, FLe studies with moral dilemmas [6,16] have suggested that a second language (L2) provides to bilingual people a greater emotional distance than their first, first language (L1), leading to more utilitarian decisions using L2 than using L1. A recent study [9] revealed that emotional concerns are in fact reduced in low-proficiency speakers, but better comprehension diminishes the FLe. However, a meta-analysis [17] revealed that proficiency does not moderate the FLe. Instead, languages sharing lexical and/or grammatical similarities did not show the FLe due to a strong cultural influence of the L1 on L2 or due to the linguistic similarities between both languages. Nevertheless, this statement is far to be conclusive because study of the FLe using close languages has been conducted mainly with Germanic languages, such as English, Dutch, German, Swedish, and Norwegian (see [18]). In contrast, the study by Miozzo et al. [19] observed that the FLe was present even in close Romance languages (Italian-Venetian, Italian-Bergamasque), in early, high proficient bilinguals.

It has been proposed (e.g., [6,16]) that a second language (L2) provides bilingual people a greater emotional distance than their first language (L1). Regarding this so-called "Disembodied cognition" [20], recent studies exploring the relation between L2, and the amount of emotion involved in a moral dilemma show mixed results. A recent study by Brouwer [2] found that L2 leads to more utilitarian decisions, even in early highly proficient Dutch-English bilinguals. However, that was true only for personal dilemmas but not for impersonal ones. This result is congruent with previous literature (e.g., [21,22]) and suggests that FLe is modulated by the amount of emotion involved. Impersonal dilemmas are emotionally distant; thus, people respond to them in the same manner regardless of the language the dilemma is presented. Conversely, personal dilemmas are perceived more emotionally when presented in L1, so L2 (less emotionally) leads to more utilitarian responses than L1. In other words, on the one hand, the L1 is more emotional, which is congruent with highly emotional personal dilemmas leading to more deontological (not do it) responses; on the other hand, L2 lets bilingual people take some distance and respond more utilitarianly. Nevertheless, it seems plausible to say that, for early and proficient bilinguals, L2 should be as emotional as L1; thus, FLe should be absent,

because L2 is not a foreign language for simultaneous early bilinguals. In fact, several studies support this idea [10,18,23], suggesting that factors such as cultural influence and linguistic similarity diminish the effect of L2 leading to more utilitarian responses.

Similarly, two studies explored the FLe in early English-Chinese bilinguals [24,25] and found no differences in utilitarian responses between the English and the Mandarin Chinese versions of the moral dilemmas. However, higher emotional arousal showed a positive relation with the number of utilitarian decisions. This result is contradictory with the DPM described by Greene and colleagues [13,26], by which high arousal would lead to a deontological response (not do it).

In this vein, several recent studies have raised concerns about the DPM, especially when it comes to the role of emotions in moral judgements [27–29]. A new model based on consequences, norms, and a general preference for inaction (CNI model) has been proposed [30]. Since sacrificial-type dilemmas present harmful actions promoting the greater good, some authors (e.g., [31]) propose that sacrificial responses reflect psychopathic traits and diminished empathy (low aversion to harm) rather than genuine utilitarianism (the lesser evil). In fact, many studies found that individuals with anti-social traits tend to give utilitarian responses (e.g., [32–38]). Other studies, however, found little or no relation between psychopathy and morality [39,40]. A recent study [41] explored this question, among others, from a third-party perspective. Results showed that psychopathic traits were associated with, on one hand, judging more punishable a killing by selfish motives than a killing by helping others; and, on the other hand, a reduced concern about the death of avoidable victims. Contrary to these results, previous research (e.g., [11,42–44]) showed that people, in general, tend to be selfish (respond more utilitarianly if they are beneficial of the harm caused) and prefer to harm others when the harm is not avoidable (for instance, when the person who is killed was going to die anyway). Moreover, from the Triarchic Model perspective [45], it would be expected that psychopathic traits like meanness were associated with selfish motives and more utilitarian decisions (e.g., [32]). Thus, Behnke's et al. [41] results could indicate that factors like benefit recipient (self-beneficial vs. other beneficial) and avoidability of the harm (avoidable vs, unavoidable death) should be considered, especially when exploring the effect of psychopathic personality. Additional research in a first-person perspective, exploring whether psychopathic traits interact with the arousal of the dilemma and other emotional variables that are known to also affect the moral response, such as valence and dominance [46,47] is thus of particular interest.

Therefore, the present study has two primary objectives. Firstly (O1), to elucidate potential reasons for the inconsistency in observing FLe in bilinguals. As past research has indicated that factors such as language proficiency, linguistic distance, contextual language use, age of L2 acquisition may influence the presence of FLe (e.g., [9,17–19]), the present research addresses these factors by recruiting a group of early highly proficient bilinguals. Notably, our study differs from prior research by examining two closely related languages (Catalan and Spanish) in a dual-language context (Mallorca). While prior research already explored cross-linguistic differences between nearby languages (e.g., [10,18,19]), our research extends this inquiry to the unique linguistic dynamics present between two closely related and co-official Romance languages. It is important to remember that Miozzo et al. [19] already studied FLe in Romance languages, but Venetian and Bergamasque are spoken in different contexts, namely households and informal circles, unlike Italian. Thus, the context in which close languages are used could be a relevant factor to explore in the FLe even when both languages are learned early in life and are linguistically close. The present study will further clarify the presence of the FLe in early high proficient bilinguals using Romance languages, immersed in a dual-language context (in contrast to single-language contexts, as noted by [48]), in which both languages are used equally in the same contexts and not being one of them restricted to informal circles. It is

worth noting that Catalan and Spanish are co-official languages present at both educational, media and social settings. Our study will add robustness to the findings related to the FLe and will better delimit the circumstances under which the FLe is observed.

Moreover, drawing from the insights of previous studies [11,32,34,36,37,40,42], dilemma-specific factors, participants' assessments of the dilemmas, and psychopathic traits will be controlled. This comprehensive approach seeks to determine whether the DPM can account for the results or if a CNI model is more appropriate [3,30,37,49–51]. This leads us to our second objective (O2), to add some novel data to moral decision-making and FLe research by exploring the individual and combined influences of psychopathic traits and dilemma variables on moral decision-making. To do so, we explore the effect of personal force, benefit recipient and avoidability, and participants' assessments of the dilemmas in terms of emotion arousal, valence, and dominance. As language can affect mental imagery [52], we additionally explore the participants' assessments of the vividness of the dilemmas (i.e., how intense and detailed the dilemma was imagined). We also assessed the perceived difficulty of making the decision as an indicator of the emotional burden and inner conflict associated with the moral choice [39]. Furthermore, we collected data on perceived verisimilitude, which gauged how closely the dilemmas resembled real-life situations. This was done in response to reported disparities between real and hypothetical moral choices [53]. Finally, interactions of previous variables with psychopathic traits like meanness, boldness, and disinhibition will also be considered.

To recapitulate, we propose the following goals and hypotheses:

1. To test predictions based on previous findings in moral judgement literature.

H1a. In general, participants will tend to respond deontologically to personal dilemmas, while responding utilitarianly to impersonal ones (e.g., Christensen et al., [42]). Personal force may also affect decision time (DT), as expected by a DPM perspective (e.g., [26]); thus, personal dilemmas will yield quicker responses than impersonal ones.

H1b. Regarding language, we expect no differences between responses to moral dilemmas presented in L2 with respect to L1, as our participants are highly proficient early bilinguals as in Wong and Ng [25] and the fact that the languages in our study are closely related and used equally in all contexts, making the traditional distinction between L1 and L2 less prominent, in contrast to Miozzo et al. [19].

H1c. Participants with high scores on psychopathy will give more utilitarian responses than those with low scores. Departing from the Triarchic Model perspective [45], we expect meanness, but not boldness or disinhibition, to be associated with a larger proportion of utilitarian decisions (e.g., [32]).

H1d. If the DPM is correct (e.g., [26]), higher arousal will lead to less utilitarian responses, and it will emerge as a modulator between dilemma type, language, or psychopathic traits, and the utilitarian response.

2. To explore how psychopathic traits, dilemma type, and participant's assessment of the dilemmas, would separately and jointly affect moral response and DT.

H2a. We expect that, in addition to the effect of personal force on the moral response, the benefit recipient and avoidability of the harm will play a role in the moral decisions (e.g., [41]).

H2b. In addition to arousal of the dilemma, the participant's assessment of valence, dominance, difficulty, verisimilitude, and vividness will also affect the moral choice and/or DT [46,52].

H2c. We hypothesize that meanness will interact with dilemma variables and/or dilemma assessments, affecting moral responses and/or DT.

## Method

### Participants

Students from the Faculty of Education of the University of the Balearic Islands who considered themselves bilingual were invited to volunteer in this study. Fifty-seven undergraduate students (44 females, 13 males; age range = 18–46 years; M = 24.12, SD = 5.06) volunteered to participate. Following our eligibility criteria, five participants were excluded because they didn't consider themselves balanced bilinguals and/or their first language was not Catalan. To avoid sample size biases, we planned to carry out a multilevel multivariate logistic regression analysis. We also assess the adequacy of the sample size based on the odds ratio obtained in previous studies examining the effect of dilemmas' variables with a similar analytical strategy (OR = 0.69; [24]). For a .80 power, OR = 0.69, and $p$ = .05, an a priori power analysis using G $^*$ Power [54] for logistic regression yielded a sample size recommendation of 730 observations. Considering that each participant responded to 16 moral dilemmas (see Materials for details) and the multilevel structure of our database where each trial constitutes an observation (see Statistical analyses), we obtained 766 valid trials from 52 participants (39 females; age range = 21–46 years; $M$ = 24.12, $SD$ = 4.94). Catalan was the familiar language and first language learned for all participants (L1), who considered themselves bilingual and learned their second language, Spanish (L2), before 3 years of age, $M$ = 2.63, $SD$ = 1.94. The overall proficiency score was computed for each language as the mean of oral and written expressive proficiency, and oral comprehension for each language. Scores were obtained by means of five-point Likert self-reported scales quantifying the competence in each of these language skills (0 = *very poor*, 1 = *poor*, 2 = *intermediate*, 3 = *good*, 4 = *very good*). Participants were highly proficient in both languages, $M$ = 3.67, $SD$ = 0.37, and $M$ = 3.25, $SD$ = 0.57 for L1 and L2 overall proficiency, respectively. However, participants were not equally proficient in both languages, $t(51)$ = 5.787, $p < .001$, $r$ = .63. It is important to note that, given the bilingual context of our participants, their proficiency distribution reflects a natural outcome of their usage patterns and linguistic exposure, rather than indicating perfectly balanced bilingualism. Demographic data regarding gender, age, and level of studies of participants can be consulted in Table 1.

No participant reported psychiatric or pharmacological treatment at the time of testing, and none presented non-corrected visual or auditory deficits. This study was approved by the

**Table 1. Descriptive statistics for sample demographic variables.**

| Variable | *n* | % |
|---|---|---|
| Gender | | |
| Male | 13 | 25,0% |
| Female | 39 | 75,0% |
| Education | | |
| 1 year of college | 4 | 7,7% |
| 2 years of college | 35 | 67,3% |
| 3 years of college | 8 | 15,4% |
| 4 years of college | 1 | 1,9% |
| 6 years of college | 4 | 7,7% |

*Note. N* = 52.

Committee on Research Ethics of our University (approval nr. 2647). All participants were informed about the nature of the study and provided written informed consent in accordance with the Declaration of Helsinki. Authors had no access to information that could identify individual participants during or after data collection.

## Materials

**Triarchic psychopathy measure.** The TriPM consists of 58 self-reported items that measure the three phenotypic domains of boldness, meanness, and disinhibition proposed in the triarchic conceptualization of psychopathy [45]. Boldness involves several key attributes: the ability to remain composed and focused during moments of pressure or danger, the capacity to quickly recover from stressful situations, strong self-confidence and social effectiveness, and tolerance for unfamiliar and risky circumstances. Meanness encompasses various traits, including a lack of empathy, a disregard for forming close connections with others, a tendency toward rebellion, seeking excitement, taking advantage of others, and deriving a sense of power through acts of cruelty. Lastly, disinhibition refers to a general phenotypic propensity towards impulse control problems, which may manifest as a lack of foresight and planning, difficulty regulating emotions and urges, a preference for immediate gratification, and limited behavioral restraint. The items are answered using a 4-point Likert scale: 1 = *true*, 2 = *somewhat true*, 3 = *somewhat false*, and 4 = *false*. We used the Spanish version of the TriPM adapted by Poy et al. [55]. Cronbach's alpha in this study was .75 for boldness, .80 for meanness, and .74 for disinhibition.

**Moral dilemmas task.** We selected a series of 16 standard moral dilemmas from the set of 46 moral dilemmas validated in Spanish by Christensen et al. [11] and translated to Catalan also by Christensen et al. [11]. Our selection obeyed to maximize the same quantity of dilemmas in three categories: personal force, benefit recipient, and avoidability of the harm (death of one person, in all the cases). So, we blocked the intentionality variable and our 16 scenarios included instrumental harm. There were no accidental death scenarios (in all our dilemmas, a person *has to* die to save other people). The dilemmas selected were eight personal moral dilemmas (PR) and eight impersonal moral dilemmas (IP). In each of these two categories, four of the eight scenarios were self-beneficial (SB) and four were other-beneficial dilemmas (OB). Finally, two of every four self/other beneficial dilemmas ended in an unavoidable death (UV) while in the others death could be avoided (AV). In addition, we included four nonmoral dilemmas (NM) from Greene et al. [14], translated to Spanish and Catalan by our team, as a way to control that participants were paying attention to the task. Two additional dilemmas (one moral and one nonmoral) were also selected for practice; one-half of the dilemmas were presented in Catalan and the other half was presented in Spanish. See the materials in our Open Science Framework website (https://www.doi.org/10.17605/OSF.IO/NZRP3) for more information about the dilemmas.

## Procedure

All participants filled out online forms of self-reported instruments, such as the language proficiency scores and the TriPM, and demographic data regarding gender, age, and level of studies (Table 1). Once all participants had answered the self-reported instruments, the experimental session with the moral dilemmas task took place in a quiet, dimly lit room, and lasted about 30 minutes. The consent form and oral instructions for the computerized tasks were presented in their L1, Catalan. The TriPM was administered in Spanish (L2) because it was already published and available in that language.

A block of 10 dilemmas in Catalan and a block of 10 dilemmas in Spanish, counterbalancing the order of language blocks across participants, were presented by E-Prime 2.0 [56] to all participants. Personal force, benefit recipient, and avoidability were also counterbalanced across languages and participants. It is important to notice that, although Catalan and Spanish dilemmas coincided in content, no participant saw the very same dilemma twice: if the personal version of the dilemma was presented in Catalan, the impersonal version of the dilemma was presented in Spanish, and vice versa. The task began with an instructions screen presented in L1, followed by the practice stimuli. Each dilemma was presented in dark grey Helvetica font, pt 16, on a pallid grey screen. Participants read at their own pace and decided if they would do the action presented by the dilemma, responding by pressing the I key (meaning "yes") or O key (meaning "no") on a Spanish QWERTY keyboard. These keys were used because the letters "i" and "o" are part of the Catalan (and Spanish) "Sí" and "No" words ("Yes" and "No"), so the keys are intuitively related to the response they represent. Moreover, these keys are located side by side in a Spanish QWERTY keyboard, allowing participants to respond comfortably with one hand. Decision time (DT) was also recorded and, to address potential implications of hand preferences for DT, we asked about participants' hand preference, and they responded with their dominant hand. Following the moral decision, participants were asked to evaluate the dilemma on three emotional scales: affective valence (from *unpleasant* to *pleasant*), activation (from *not arousing* to *very arousing*), and dominance (from *no control over the situation* to *full control over the situation*). For these three scales, participants were instructed to use a 9-point Self-Assessment Manikin (SAM) Likert scale [57] and respond by pressing numbers 1–9 on the keyboard. In addition, they assessed every situation on the scales of difficulty (how hard it was to respond to the dilemma), verisimilitude (how much did the dilemma seem to be a real situation), and vividness (how much intense and detailed the dilemma was imagined). For these three scales, they were instructed to use a 9-point Likert scale. Fig 1 illustrates the procedure. See our OSF website (https://www.doi.org/10.17605/OSF.IO/NZRP3)for more information about the procedure, dilemmas, and assessment scales.

The study was carried out during the year 2015 in a laboratory of the university suited for experiments. Participants carried out the task individually. The viewing distance was approximately 16 inches from the screen.

## Statistical analyses

All analyses were performed by means of IBM SPSS Statistics 28.0 [58]. Descriptive statistics were obtained for the self-reported measures of the study (L1 and L2 proficiency, TriPM measures for boldness, meanness, and disinhibition) and for measures from the moral dilemma task. Responses to moral dilemmas with DT < 1000 ms (2.4% of trials) were excluded from the analyses. Outlier DT values were also excluded (jackknife distances > 4, 5.5% of trials).

Only the 16 moral dilemmas were included in the statistical analyses. The data had a multi-level structure with within-subjects dilemma variables (Personal Force, Benefit Recipient, Avoidability, and Language of presentation) and between-subjects participant variables (Boldness, Meanness, and Disinhibition Scores). Utilitarian Response (UR), where 1 indicates utilitarian response, and 0 indicates a deontological response; and DT were considered as the dependent variables. In total, there were 766 valid trials collected from 52 participants. A generalized linear model approach was adopted to handle the multilevel data by means of Generalized Estimating Equations (GEE, binomial distribution, logit link, robust estimation). Tables 2 and 3 show descriptive statistics of the main variables and covariates.

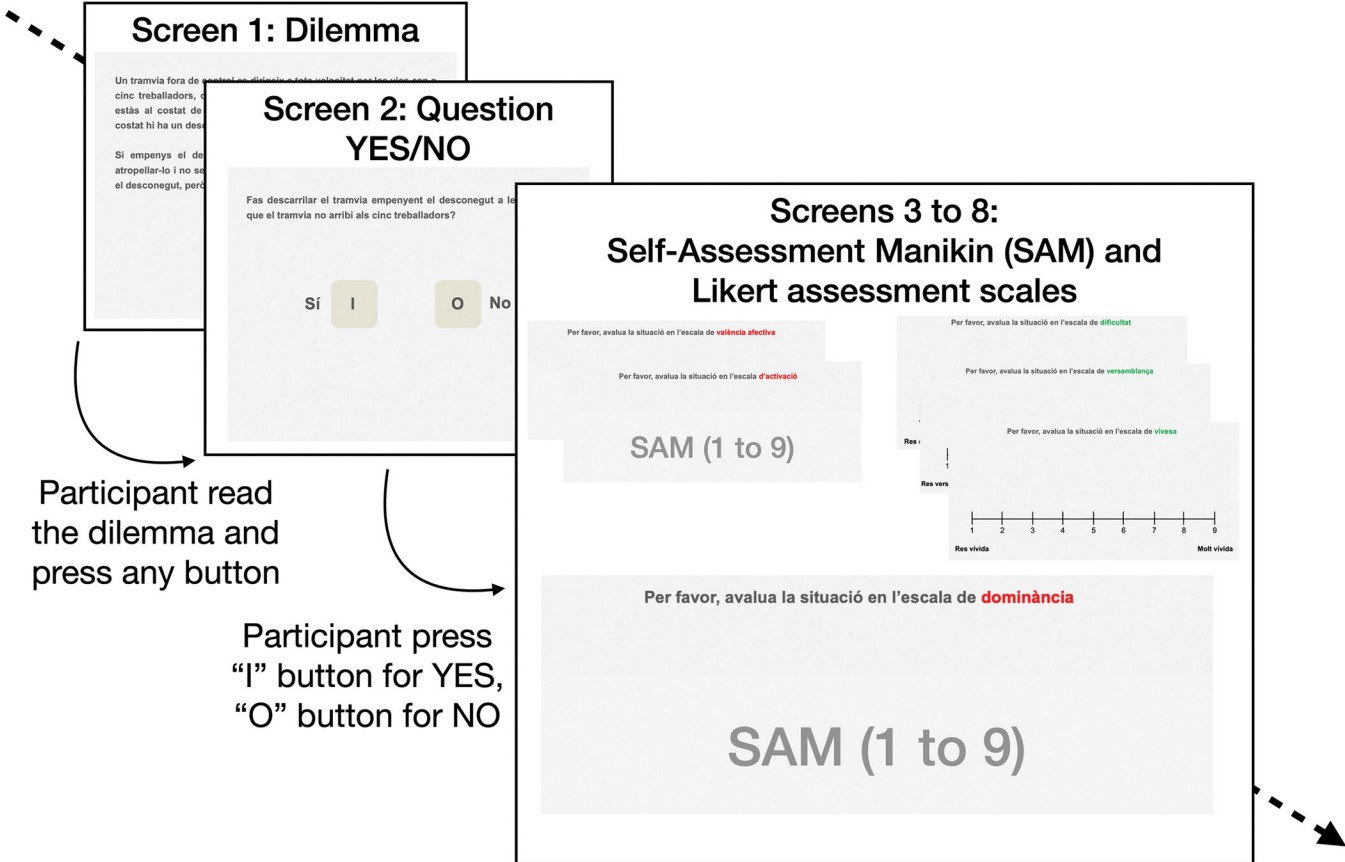

**Fig 1. Example of experimental procedure.** Screenshots are taken from the practice trials (in Catalan -L1-). Each of the two blocks (L1, L2, counterbalanced) consisted of two non-moral dilemmas (different than the ones in the other block) included for controlling participant's engagement in the task, four personal moral dilemmas (shown in their impersonal version in the other block) and four impersonal moral dilemmas (shown in their personal version in the other block). Participants read each dilemma at their own pace, pressing a button to move from one screen to the next.

To test hypotheses H1a and H1b, each dependent variable was regressed on two predictors, namely (i) personal force (PR, personal dilemma, and IP, impersonal dilemma), and (ii) language of presentation (L1, Catalan, and L2, Spanish). Two covariates, (i) arousal of the dilemma (assessed by every participant), and (ii) difference in language proficiency (L1-L2) of each participant, were controlled for. To test H1c, each dependent variable was regressed on three predictors, which were the three scales from TriPM, (i) boldness, (ii) meanness, and (iii) disinhibition. One covariate, arousal of the dilemma assessed by the participant, was also included in the model. To test H1d, arousal of the dilemma was included in the model of previous analyses, as stated above.

To test H2a, each dependent variable was regressed on two predictors from the dilemmas: (i) benefit recipient (SB, self-beneficial, and OB, other beneficial) and (ii) avoidability of the harm (AV, avoidable, and UV, unavoidable). Personal force (PR and IP) was also included in the model.

To test H2b, each dependent variable was regressed on five predictors from the participant's assessment of the dilemmas: (i) valence, (ii) dominance, (iii) difficulty, (iv) verisimilitude, and (v) vividness. Arousal was also included in the model.

**Table 2. Descriptive statistics for all variables collected as a function of the type of moral dilemma.**

| Dilemma variables | UR proportion | | DT in milliseconds | | Valence | | Arousal | | Dominance | | Difficulty | | Verisimilitude | | Vividness | |
|---|---|---|---|---|---|---|---|---|---|---|---|---|---|---|---|---|
| | *M* | *(SD)* | *M* | *(SD)* | *M* | *(SD)* | *M* | *(SD)* | *M* | *(SD)* | <u>*M*</u> | *(SD)* | *M* | *(SD)* | *M* | *(SD)* |
| Language | | | | | | | | | | | | | | | | |
| Catalan (L1) | .46 | (.50) | 7892 | (5189) | 2.07 | (1.75) | 7.28 | (1.77) | 5.20 | (2.78) | 7.08 | (2.37) | 4.69 | (2.55) | 6.07 | (2.52) |
| Spanish (L2) | .46 | (.50) | 8215 | (4997) | 2.02 | (1.75) | 7.23 | (1.72) | 5.36 | (2.80) | 7.14 | (2.37) | 4.79 | (2.55) | 6.11 | (2.45) |
| Personal force | | | | | | | | | | | | | | | | |
| Personal | .39 | (.49) | 7411 | (4692) | 2.00 | (1.74) | 7.26 | (1.80) | 5.31 | (2.83) | 7.13 | (2.34) | 4.70 | (2.57) | 6.23 | (2.48) |
| Impersonal | .53 | (.50) | 8716 | (5397) | 2.09 | (1.75) | 7.25 | (1.70) | 5.25 | (2.76) | 7.09 | (2.30) | 4.78 | (2.53) | 5.96 | (2.48) |
| Benefit recipient | | | | | | | | | | | | | | | | |
| Self-beneficial | .50 | (.50) | 7627 | (4608) | 1.82 | (1.56) | 7.37 | (1.79) | 5.09 | (2.89) | 7.30 | (2.29) | 4.67 | (2.55) | 6.17 | (2.47) |
| Other-beneficial | .42 | (.49) | 8494 | (5514) | 2.27 | (1.90) | 7.12 | (1.69) | 5.48 | (2.68) | 6.92 | (2.34) | 4.80 | (2.55) | 6.02 | (2.50) |
| Avoidability | | | | | | | | | | | | | | | | |
| Avoidable | .34 | (.47) | 7736 | (4611) | 2.12 | (1.84) | 7.24 | (1.72) | 5.28 | (2.83) | 6.92 | (2.43) | 4.84 | (2.54) | 6.19 | (2.43) |
| Unavoidable | .58 | (.49) | 8380 | (5523) | 1.97 | (1.64) | 7.26 | (1.77) | 5.29 | (2.75) | 7.30 | (2.19) | 4.63 | (2.56) | 5.99 | (2.53) |

*Note.* UR: Utilitarian response ("Yes" response to the dilemma); DT (decision time in ms); L1: First language; L2: Second language

To test H2c, the previous two analyses were replicated with boldness, meanness, and disinhibition as covariables.

## Transparency and openness

We reported above the characteristics of our sample, all data exclusions, all manipulations, and all measures in the study, following JARS [59]. This study's design and its analysis were not preregistered because the data acquisition was conducted in 2016, prior to the expansion of preregistration platforms.

## Results

### Personal force (H1a) and language of the dilemma (H1b)

On the one hand, regarding the model with the predictors personal force and language, personal force (PR, IP) significantly predicted utilitarian response, ($\beta$ = .569, $p$ < .001, OR = 1.77, OR 95% CI [1.36, 2.29]), showing that impersonal dilemmas yielded 77% more utilitarian

**Table 3. Descriptive statistics for the Triarchic Psychopathy Measure (TriPM) and difference in language proficiency scores.**

| | *M* | *(SD)* | Min. | Max. |
|---|---|---|---|---|
| TriPM scale | | | | |
| Boldness | 25.65 | (7.17) | 9 | 43 |
| Meanness | 8.31 | (5.60) | 0 | 26 |
| Disinhibition | 14.65 | (5.93) | 6 | 33 |
| L1/L2 difference | 1.25 | (1.56) | -1[a] | 5 |

*Note.* L1: First language (Catalan); L2: Second language (Spanish).

[a] One participant answered that he/she was slightly more proficient in L2. It seems a mistake but some early bilinguals in Mallorca consider that they are equally or more proficient in L2 because they might feel more comfortable in written L2 as compared to L1.

responses. Language of the dilemma had no effect on moral responses, ($\beta$ = −.039, $p$ = .709, OR = 0.96, OR 95% CIs [0.78, 1.18]), indicating that utilitarian responses were not influenced by the language of the dilemma. Neither arousal ($B$ = −0.126, $\beta$ = −.220, $p$ = .113, OR = 0.80, OR 95% CI [0.61, 1.05]) nor the difference in language proficiency ($B$ = 0.092, $\beta$ = .140, $p$ = .382, OR = 1.15, OR 95% CI [0.84, 1.57]) affected the response either. There were no significant interactions (all $ps$ > .288).

The same analysis approach with DT as the dependent variable yielded similar results, with personal force as the only significant main effect, ($B$ = 1.32, $\beta$ = 0.26, $p$ < .001, OR = 1.30, OR 95% CI [1.13, 1.48]), indicating that impersonal dilemmas were responded slower than personal ones. No other effects and no interactions reached significance (all $ps$ > .13).

## Psychopathic traits (H1c) and arousal (H1d)

On the other hand, in regard to the model with the dimensions of the TriPM as predictors, meanness significantly predicted UR, ($B$ = 0.098, $\beta$ = .555 $p$ = .008, OR = 1.74, OR 95% CI [1.16, 2.62]), indicating that the higher the meanness score, the higher the probability to respond utilitarianly. Boldness ($B$ = 0.008, $\beta$ = .055, $p$ = .662, OR = 1.06, OR 95% CI [0.82, 1.35]) and disinhibition ($B$ = −0.033, $\beta$ = −.194, $p$ = .234, OR = 0.82, OR 95% CI [0.60, 1.13]) were neither significant predictors of UR. Arousal did not affect the proportion UR ($B$ = −0.084, $\beta$ = −.147, $p$ = .227, OR = 0.86, OR 95% CI [0.68, 1.10]). However, the interaction between meanness and arousal was significant ($B$ = 0.048, $\beta$ = .474, $p$ = .016, OR = 1.61, OR 95% CI [1.10, 2.36]) being the effect of arousal on meanness ($B$ = −0.036, $\beta$ = .327, OR = 1.38) an indicator that the odds of meanness predicting UR increases about 38% for every unit increased in arousal. Fig 2 illustrates this interaction.

The same design with DT as the dependent variable yielded no significant results. No psychopathic dimension predicted DT (all $ps$ > .088).

## Benefit recipient and avoidability (H2a)

Fig 3 shows how dilemma types affect the UR proportion. In terms of benefit recipient and avoidability, both were significant predictors of UR. Self-beneficial dilemmas predicted a larger number of UR, ($\beta$ = .324, $p$ = .04, OR = 1.38, OR 95% CI [1.02, 1.88]); while unavoidable dilemmas were also associated with more UR ($\beta$ = .99, $p$ < .001, OR = 2.70, OR 95% CI [2.06, 3.54]). There was a significant interaction between benefit recipient and avoidability, ($\beta$ = −0.59, $p$ = .02, OR = 0.55, OR 95% CI [0.34, 0.91]). The effect of unavoidable dilemmas when they were also self-beneficial ($\beta$ = 0.538, OR = 1.71) indicates that self-beneficial dilemmas that narrate unavoidable harm have nearly twice the odds of being responded utilitarianly than the other types of dilemmas.

The same analysis with DT as the dependent variable yielded similar results. Self-beneficial dilemmas predicted a decrease in DT, ($B$ = −0.88, $\beta$ = −.86, $p$ < .001, OR = 0.92, OR 95% CI [0.87, 0.96]), showing that people respond to self-beneficial dilemmas quicker than other-beneficial ones. There also was a significant interaction between benefit recipient and avoidability, ($B$ = −3.26, $\beta$ = −.160, $p$ < .001, OR = 0.85, OR 95% CI [0.80, 0.90]). The effect of unavoidable dilemmas when they were also self-beneficial ($B$ = −1.23, $\beta$ = .46, OR = 1.58) indicates that participants responded 58% quicker when self-beneficial dilemmas would finish with unavoidable harm.

## Participant's assessments of the dilemmas (H2b)

The regression model including participant's assessments of the dilemmas as predictors revealed dominance ($B$ = 0.185, $\beta$ = .517, $p$ < .001, OR = 1.68, OR 95% CI [1.23, 2,28]) and difficulty ($B$ = 0.195, $\beta$ = .453, $p$ = .03, OR = 1.57, OR 95% CI [1.05, 2.36]) as the only significant

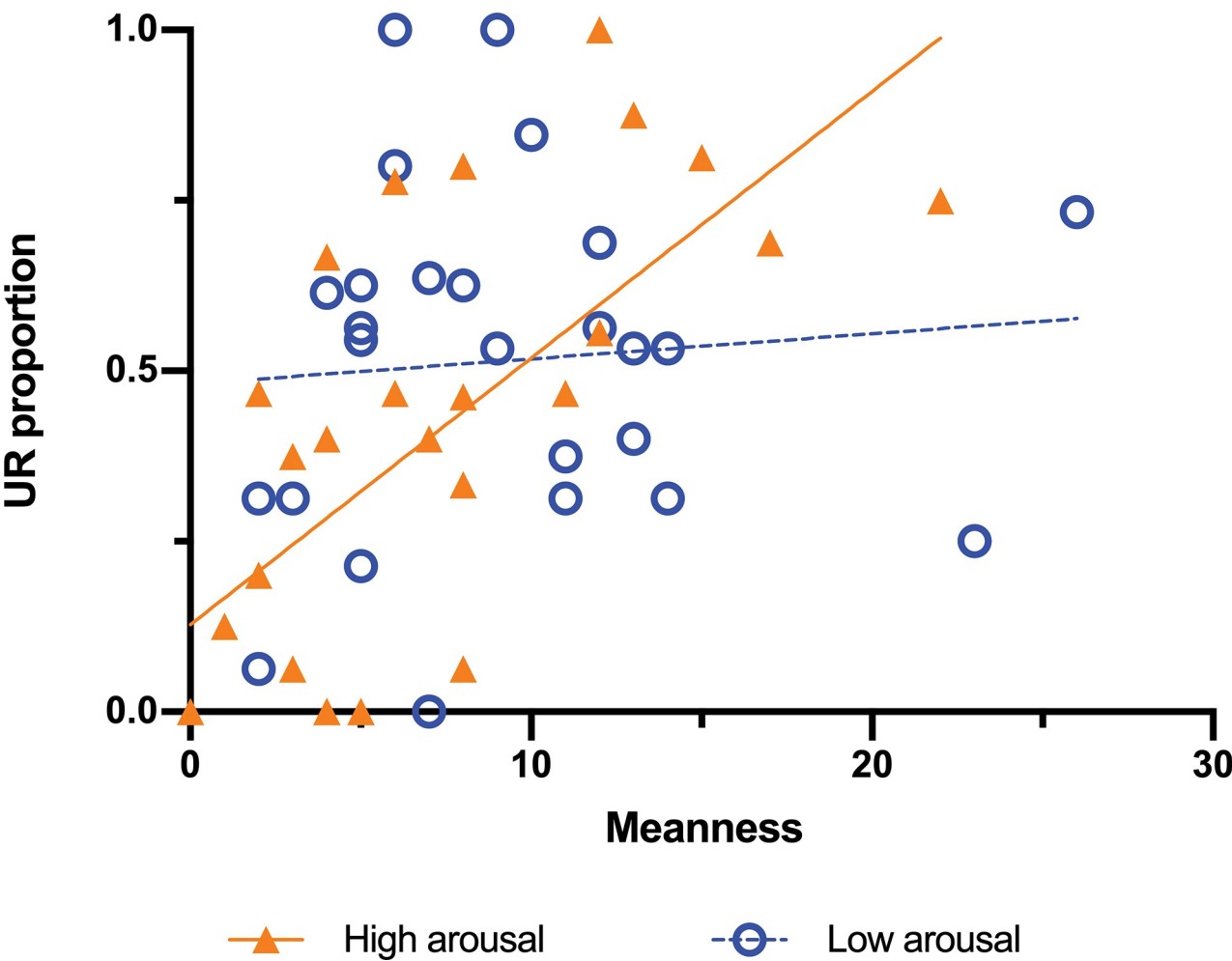

**Fig 2. Correlation between meanness and the proportion of UR.** Meanness and UR proportion were significantly correlated ($r = .403$, $p = .001$). Here the arousal variable was dichotomized based on its median, illustrating that the correlation of Meanness with UR is stronger for higher levels ($r = .654$, $p < .001$) than for low levels of arousal ($r = .084$, $p = .682$). The fitted regression model for high arousal was: UR proportion = 0.03912*Meanness + 0.1276.

predictors of UR. According to these results, people tend to respond more utilitarianly not only when they perceive more control over the situation (dominance) but also when the decision is perceived as difficult. Only valence interacted significantly with arousal, ($B = 0.83$, $\beta = .253$, $p = .037$, OR = 1.29, OR 95% CI [1.01, 1.63]). The effect of valence on arousal ($B = 5.084$, $\beta = .284$, OR = 1.32) indicates that dilemmas perceived as arousing were more likely to be responded utilitarianly where these were evaluated with a less aversive valence.

The same regression model with DT as the dependent variable yielded difficulty, ($B = 0.218$, $\beta = .99$, $p = .009$, OR = 1.10, OR 95% CI [1.02, 1.19]), and vividness, ($B = 0.270$, $\beta = .13$, $p = .002$, OR = 1.14, OR 95% CI [1.05, 1.24]), as significant predictors; thus, dilemmas perceived as difficult and vivid led to slower DT. No interactions reached significance (all $ps > .076$).

### Psychopathic traits as covariables (H2c)

Finally, when psychopathic traits were included in the model together with personal force, benefit recipient, and avoidability as predictors of UR, no interactions were significant.

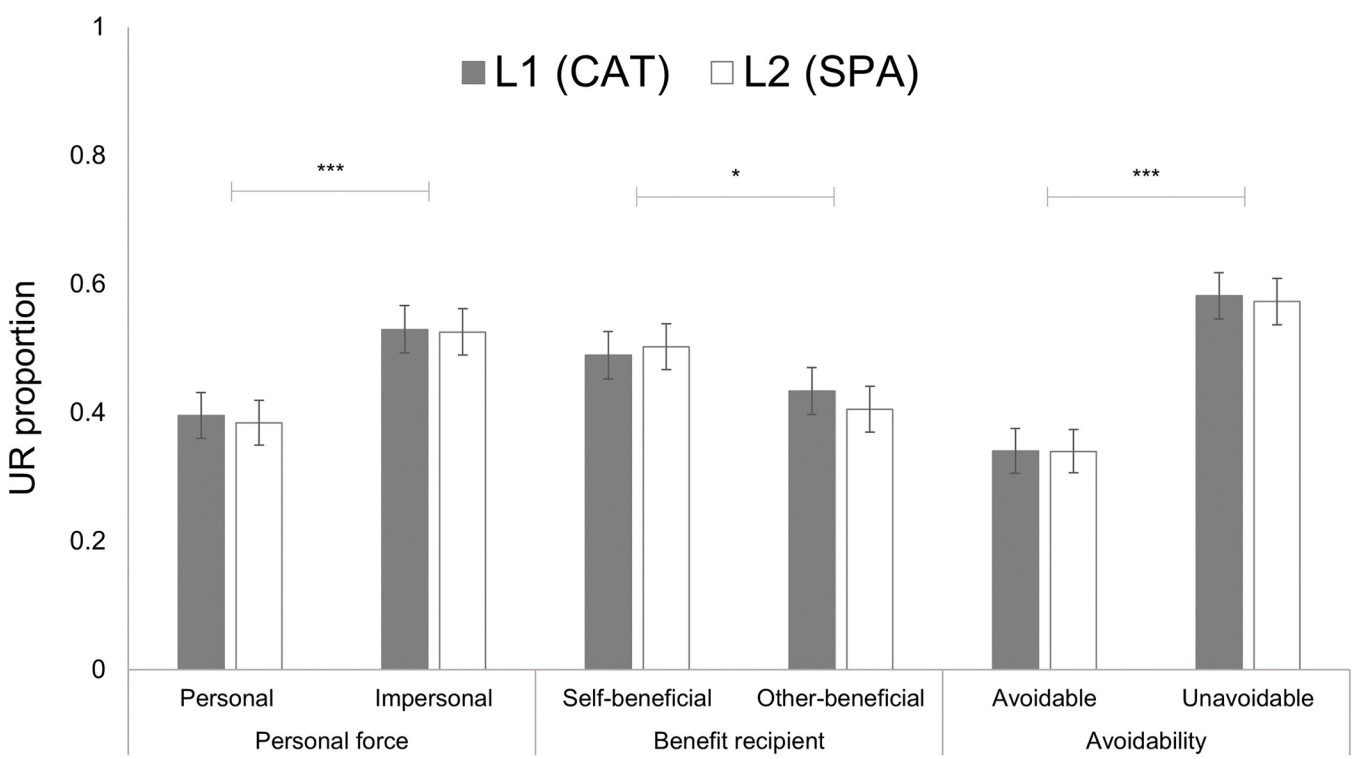

**Fig 3. Mean proportion of UR as a function of language, personal force, benefit recipient, and avoidability.** Mean UR proportion was significantly greater for the following: impersonal compared to personal, self-beneficial compared to other-beneficial, and unavoidable compared to avoidable dilemmas. No significant effect was found for language. Error bars show standard errors. *$p < .05$, **$p < .01$, ***$p < .001$.

However, in regard to the regression model including the participant's assessments of the dilemmas as predictors, the interaction between meanness and vividness was significant, ($B = 0.025$, $\beta = .356$, $p = .049$, OR = 1.43, OR 95% CI [1.00, 2.02]). This indicated that, the more vivid the dilemmas were perceived, the more the odds to be responded utilitarianly for people scoring higher in meanness. Boldness significantly interacted with dominance, ($B = -0.016$, $\beta = -.326$, $p = .028$, OR = 0.72, OR 95% CI [0.54, 0.97]). The effect of dominance on boldness was ($B = 0.141$, $\beta = .251$, OR = 1.28), indicating that, when participants perceived that they dominated the situation, dilemmas were more likely to be responded utilitarianly as a function of the participant's higher boldness.

In regard to the regression model with personal force, benefit recipient, and avoidability as predictors of DT, three interactions reached significance when we included psychopathic traits as covariables. First, personal dilemmas interacted with meanness, ($B = -0.100$, $\beta = -.06$, $p = .028$, OR = 0.95, OR 95% CI [0.90, 0.99]). The effect of personal force on meanness ($B = -0.318$, $\beta = -.43$, OR = 0.65) indicates that participants respond about 35% quicker to personal dilemmas for every unit increase in meanness. Second, personal force interacted with disinhibition ($B = 0.138$, $\beta = .08$, $p = .008$, OR = 1.08, OR 95% CI [1.02, 1.15]). The effect of personal force on disinhibition ($B = -0.08$, $\beta = .21$, OR = 1.23) indicates that participants respond about 23% quicker to personal dilemmas for every unit increased on disinhibition. Finally, the benefit recipient interacted with boldness, ($B = 0.073$, $\beta = .051$, $p = .031$, OR = 1.05, OR 95% CI [1.00, 1.10]). The effect of boldness on the benefit recipient ($B = -0.023$, $\beta = -.037$, OR = 0.96) indicates that, with larger scores in boldness, participants tend to respond quicker to self-beneficial dilemmas.

Regarding the regression model with participants' assessments as predictors of DT, when we included psychopathic traits as covariables, four interactions reached significance. First, boldness interacted with verisimilitude, ($B = 0.034$, $\beta = .120$, $p = .02$, OR = 1.13, OR 95% CI [1.02, 1.24]). The effect of verisimilitude on boldness ($B = 0.173$, $\beta = .059$, OR = 1.06) indicates that the odds for boldness score predict DT increase as more verisimilar is a dilemma. Second, boldness also interacted with vividness, ($B = -0.046$, $\beta = -.159$, $p = .002$, OR = 0.85, OR 95% CI [0.77, 0.94]). The effect of boldness for vividness ($B = 0.093$, $\beta = -.22$, OR = 0.80) indicates that the odds of responding faster to vivid dilemmas increase about 20% for every unit increase of boldness. Third, disinhibition interacted with vividness, ($B = -0.036$, $\beta = -.104$, $p = .042$, OR = 0.90, 95% CI [0.81, 1.00]). The effect of disinhibition for vividness ($B = -0.065$, $\beta = -.165$, OR = 0.84) indicates that the odds of responding faster to vivid dilemmas slightly increase by about 16% for every unit increase in disinhibition. Finally, dominance interacted with vividness, ($B = -0.069$, $\beta = -.094$, $p = .035$, OR = 0.91, OR 95% CI [0.83, 0.99]). The effect of dominance for vividness ($B = 0.478$, $\beta = -.158$, OR = 0.85) indicated that the odds of responding faster to vivid dilemmas increase by about 15% for every unit increase in dominance.

## Additional analysis: Predictors of utilitarian responses

In order to provide a more comprehensive examination of the studied factors and their possible interactions, a model that included the variables that best predicted UR was created, i.e., personal force, meanness, benefit recipient, and avoidability, to identify significant interactions. The model yielded no significant interactions.

As a final exploratory analysis, we regressed each dependent variable with the other. DT slightly significantly predicted UR, ($\beta = -.033$, $p = .049$, OR = .96, OR 95% CI [.94, 1.00]) and UR predicted DT, ($\beta = .86$, $p = .049$, OR = 2.36, OR 95% CI [1.00, 5.57]). It is worth noting that participants dedicated 136% more time to respond utilitarianly than to respond deontologically.

## Discussion

The present study investigated the so-called FLe in early bilinguals and the relation of psychopathic traits and emotional arousal with the number of utilitarian answers and decision times when facing moral dilemmas. We expected to replicate previous findings reported in the literature in our sample of Catalan-Spanish highly proficient bilinguals. We hypothesized that personal dilemmas would be responded quicker and more deontologically than impersonal ones (H1a), that there would be no differences according to the language of the dilemma (H1b), that participants scoring higher in meanness would respond more utilitarianly (H1c), and that arousal would modulate the relations between these variables and the response (H1d).

All our departing predictions were confirmed, although arousal only interacted significantly with meanness and not with the other variables. We also planned to explore how dilemma type beyond personal force (H2a), participant's assessments of the dilemmas beyond arousal (H2b), and psychopathic traits (H2c) would affect moral responses provided and decision time. We found that self-beneficial and unavoidable dilemmas have twice the odds of being responded quicker and utilitarianly. Regarding participant's assessments of the dilemmas, dominance, and difficulty were the best predictors of the response given, while difficulty and vividness were the best predictors for the decision time. Psychopathic traits interacted with personal force and benefit recipient variables; these traits also interacted with certain participants' assessments, specifically with dominance, verisimilitude, and vividness. Finally, we found that dominance also interacted with vividness. We will discuss these results below.

As expected (H1a), we found that, in general, participants tend to respond more utilitarianly to impersonal dilemmas than to personal ones. Indeed, according to the OR, it is 77% more probable to respond utilitarianly to an impersonal moral dilemma than to a personal one. Additionally, personal force influenced DT, and utilitarian responses to impersonal dilemmas were slower than responses to personal ones. These results are congruent with previous literature [11,14,15,26,60]. Personal dilemmas, in which participants evaluate an action that directly causes the death of another person, are more emotional and tend to be responded to more deontologically (*I would not do it*) than impersonal dilemmas, in which some mechanism mediates the death, by which they are emotionally more distant, and take longer to be evaluated.

Previous studies with bilinguals have shown how L2 could be distancing participants from the action, perhaps because L2 "disembodies" the affect of the situation [20,61]. This might lead bilingual people to respond more utilitarianly in their second language [6,16,21]. As previously stated, this so-called FLe (see [17] for a meta-analysis) seems to be present even in early bilinguals [2], although it would be plausible and congruent with previous studies, that early bilinguals do not present the FLe [10,18,19,23–25]. In the context of our investigation, it is noteworthy to consider the nature of our language pair, Spanish and Catalan, which are Romance languages that have a high degree of linguistic overlap (76% of cognate words [62]). While it is acknowledged that our findings may not encompass the full range of cross-linguistic differences observed in more disparate language pairs studied in FLe research, we contend that our exploration contributes to the broader comprehension of bilingual cognition by addressing a less frequently examined linguistic scenario. Notably, prior research has explored cross-linguistic differences between nearby languages (e.g., [18,20,22]), suggesting that linguistic similarities diminish the FLe. However, when close language pairs are restricted to a particular context, the FLe seems to be present [19]. We found no evidence of the FLe in our sample of early Catalan-Spanish bilinguals immersed in a dual-language context, as we had hypothesized (H1b), even when we controlled for differences in the participants' proficiency using L2 with respect to their L1.

Moreover, contrary to Wong and Ng's [25] study, that obtained that higher arousal leads to more utilitarian responses, contradicting the Dual Process Model (DPM) by Greene and colleagues (e.g., [13,26]), our results show no arousal interaction with personal force. Instead, our results support this model hinge on the fact that utilitarian responses spent about a third of extra DT. Participants dedicated 136% more time to respond utilitarianly than to respond deontologically. Nevertheless, if the DPM was correct, it would be expected that arousing dilemmas yielded fewer utilitarian responses (H1d). Our results did not show this effect. The arousal of the dilemma did not predict the response given. On the contrary, the new model based on Consequences, Norms, and preference for Inaction (CNI) has proved to be useful in recent years (e.g., [3,30,63]), supporting the idea that the tendency to commit sacrificial responses is more linked to psychopathic traits than a genuine utilitarianism (e.g., [31,32]). Other studies contradicted this idea, showing little or no relation between psychopathy and tendency to utilitarianism [39,40], and showing even less selfish motives in psychopaths, although they showed a reduced concern about avoidable deaths [41]. Contrary, we did find that people with high scores in psychopathic traits tend to respond more utilitarianly to moral dilemmas (H1c).

Specifically, the odds of responding utilitarianly increase by about 75% for each unit increase in meanness. This result is congruent with previous literature [32,64], and we think it complements Behnke and colleagues' study [41] because our design has a first-person perspective instead of their third-person perspective. However, our experimental design has also its limitations, and we cannot compare the two perspectives, an issue that could be addressed in

future research. Instead, our study let us find an interesting interaction of meanness with the arousal of the dilemma, as expected (H1d). Specifically, for low arousal, meanness predicts utilitarian responses but, as arousal increases the number of utilitarian responses decreases. Fig 2 illustrates this interaction. This result indicates that taking the arousal elicited by the dilemma into account is crucial to obtain more accurate results in studies of moral and psychopathic traits when it comes to utilitarian responses. Surprisingly, neither psychopathic traits nor arousal predicted the amount of time consumed before responding to the moral dilemmas, and as stated above, arousal did not modulate the effect of other predictors, such as personal force, avoidability, and benefit recipient. These variables have been shown to have an important effect on moral choices and could be interacting with participant's assessments of the dilemmas, which also play a role in the decision [11,46,65].

This leads us to our second objective, to explore how all the variables separately and jointly affect moral decisions. Regarding benefit recipient and avoidability, our results support the hypothesis (H2a) that both variables play a role in moral decisions, as reported in previous literature [11,44,66]. Specifically, self-beneficial and unavoidable dilemmas lead to an increase in the number of utilitarian responses. Moreover, dilemmas that narrate self-beneficial situations related to an unavoidable death had nearly twice the odds of being responded utilitarianly than the other types of dilemmas. In regard to the DT, participants also responded quicker to this kind of situation. This indicates that benefit recipient and avoidability jointly have a similar effect on moral choice than personal force. Interestingly, there were no interactions with personal force, despite the expectation that these factors might contribute to variations in the degree of conflict within personal dilemmas, which are traditionally considered to involve high conflict. While our approach may have limitations in representing the full spectrum of high and low-conflict dilemmas, future studies on moral decisions should not underestimate the effects of benefit recipient, and avoidability of death, taking these variables into account when studying the impact of personal variables, like psychopathic traits [41,67]. It is also noteworthy to remember that our design blocked intentionality in instrumental harm (it could be the reason why arousal and valence are very similar between the personal and impersonal versions of dilemmas). Thus, intentionality could be another key factor in varying the degree of conflict of the dilemmas and future studies should address this variable as well.

Other predictors that should continue to be explored are certain participant's assessments of the dilemmas. Our data indicate that people tend to respond more utilitarianly when they perceive more control over a particular situation (dominance) but also when the decision is perceived as difficult. Although these results may seem counterintuitive, they make sense because when participants perceived more emotional control over a situation, they engage in more extensive cognitive processing, resulting in a more difficult decision, resulting in a higher likelihood of utilitarian responses. This result is also congruent with previous literature [46,47] and aligns with DPM. Moreover, although our data did not show any effect on the responses given from either the arousal or the emotional valence of the dilemmas, there was a significant interaction between valence and arousal. Dilemmas assessed with a less negative valence (pleasantness) were more likely to be responded utilitarianly if they were also arousing. This result could complement previous findings that linked not only pleasantness but also unpleasantness with utilitarian decisions [47]. As we have stated before, from a DPM point of view [13], it would be expected that an unpleasant emotion leads to a deontological (non-utilitarian) decision. Carmona-Perera and colleagues [47] found that both a high experience of unpleasantness and a high experience of pleasantness lead to a utilitarian decision. Our results show no significant relation between valence and moral decision but, for low arousal, unpleasantness was related to a non-utilitarian response, while for high arousal, pleasantness was related to an utilitarian response style. The lack of a main effect of valence, arousal, and/or dominance

perhaps is due to limitations in our experimental design. However, our results could be supported by the CNI model [3,30], by which personal traits and thinking styles have a large impact on moral decisions.

In this vein, our exploratory analysis shows that psychopathic traits interact with participants' assessments of the dilemmas (H2c). On the one hand, although difficulty and vividness of the dilemma seem to be the assessments that best predict DT (the more vivid and difficult a dilemma is perceived, the slower the DT, as these factors are related with additional cognitive resources during the decision-making process), boldness and disinhibition also interact with vividness to predict DT. In this vein, participants tend to spend less time responding when facing vivid dilemmas as more boldness they show. Similarly, the odds of responding faster to vivid dilemmas slightly increase for each unit increase in disinhibition. Boldness also interacts with the verisimilitude assessment of the dilemma. The odds for boldness score predict faster DT increase as more verisimilar is a dilemma. These results complement previous literature that found an effect of boldness [64] and disinhibition [64,68,69] on moral choice. On the other hand, meanness and boldness also interact with certain assessments, affecting participants' responses. Specifically, people who score high in meanness tend to respond more utilitarianly as more vividly they perceive the dilemma. Moreover, boldness interacts with the dominance perceived: the higher the participant's boldness, the larger number of utilitarian decisions they give to dilemmas in which participants perceive to have control over. These results about meanness and boldness are congruent with previous literature [32,64] and complement it. Regarding meanness, when a dilemma is perceived as more vivid, it tends to evoke higher inner conflict, leading to a deontological response. However, it is only true for people scoring low in meanness. As more meanness, more utilitarian responses. It reinforces the role of meanness as a key predictor of moral choices. As for boldness, this tendency to be in control of any situation is unavoidably related to perceived emotional control (dominance) in a specific situation. More control tends to lead to more utilitarian decisions. It is worth noting here that in our regression model of DT including psychopathic traits and participants' assessments of the dilemmas, dominance interacted with vividness. The odds of responding faster to vivid dilemmas slightly increase for every unit increase of dominance; we think this highlights the importance of perceived control over a situation to evaluate cost-benefit outcomes and to decide if one follows or not the own tendency to action or inaction [37].

Finally, boldness, meanness, and disinhibition interact with the type of dilemma to predict DT (H2c). Regarding boldness, its interaction with the benefit recipient has an impact on DT. Participants scoring high in boldness seem to be faster when their own life is at risk. In regard to meanness and disinhibition, their interactions with personal force influence DT. Participants showing more meanness respond quicker to personal dilemmas, and the same holds for participants scoring high in disinhibition. These results complement previous research showing that psychopathic traits are related to the tendency to respond utilitarianly to moral dilemmas [32–38,64].

Jointly, these results lead us to two conclusions. First, early Catalan-Spanish bilinguals do not show differences when responding to moral dilemmas in L2 vs. L1. It seems plausible that, as shown in previous studies (e.g., [18]), the so-called FLe is restricted to late non-proficient bilinguals (e.g., [9]) or certain bilingual contexts (e.g., [2]). Instead, the most relevant predictor of the response seems to be the personal force of the dilemma (e.g., [11]). Second, the effects of psychopathic traits on the response to the dilemmas support the idea that explaining models for moral decision-making may involve personality variables, such as psychopathic traits (e.g., [37]), without underestimating the interaction of these variables with the type of dilemmas and, more specifically, with participants' perceptions of the dilemmas (e.g., [46,47]). However, it must be remembered that our study has some limitations, such as the use of a single language

pair, the use of only one self-reported scales to assess language proficiency and dominance, and the small sample size, which could affect the generalizability of our findings. Regarding the sample, it should also be noted that our study comprised a relatively wide age range, including two participants over 40 years of age. McNair's et al. study [70] reported age effects on moral judgment, with older adults ($> 55$ years) making significantly more deontological moral judgments. Attending to that, a planned systematic examination of potential age effects on the FLe would be reasonable for future research. Consequently, forthcoming studies could address these limitations by using larger and more diverse bilingual samples assessed with objective measures of language proficiency and deepen the research on other factors that could affect moral decision-making, such as age, cultural and educational background or alternative response options (like a gradation instead of a dichotomous answer) [10,71].

In summary, our study yielded no evidence of the FLe among early Catalan-Spanish bilinguals, suggesting that linguistic proximity may mitigate this phenomenon in dual-language contexts. Instead of language, our investigation unveiled that psychopathic traits, particularly meanness, exert a significant influence on moral decision-making. Furthermore, our findings cannot be explained by DPM or the CNI model alone, revealing distinct effects of dilemma-related variables, and the interaction of personality traits with participants' assessments of the dilemmas on moral decision-making. These results underscore the importance of considering individual differences and contextual factors when investigating moral decision-making and may contribute to the refinement of models and theories in the fields of moral cognition and psychopathy.

## Acknowledgments

We would like to thank Víctor Sánchez-Azanza and Lucía Buil-Legaz for their participation in the initial design and data collection. We also want to thank the reviewers of the manuscript because their insightful comments have significantly contributed to the refinement of our paper.

## Author Contributions

**Conceptualization:** Albert Flexas, Raúl López-Penadés.

**Formal analysis:** Albert Flexas, Raúl López-Penadés.

**Funding acquisition:** Eva Aguilar-Mediavilla, Daniel Adrover-Roig.

**Resources:** Albert Flexas, Raúl López-Penadés.

**Supervision:** Eva Aguilar-Mediavilla, Daniel Adrover-Roig.

**Validation:** Eva Aguilar-Mediavilla, Daniel Adrover-Roig.

**Writing – original draft:** Albert Flexas, Raúl López-Penadés.

**Writing – review & editing:** Albert Flexas, Raúl López-Penadés, Eva Aguilar-Mediavilla, Daniel Adrover-Roig.

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
