## [Decision Letter · Decision Letter 0]

27 Jul 2023

PONE-D-23-10940Meanness trumps language: Lack of foreign language effect in early bilinguals' moral judgmentsPLOS ONE

Dear Dr. Flexas,

Thank you for submitting your manuscript to PLOS ONE. After careful consideration, we feel that it has merit but does not fully meet PLOS ONE’s publication criteria as it currently stands. Therefore, we invite you to submit a revised version of the manuscript that addresses the points raised during the review process.

ACADEMIC EDITOR: Please revise according to the reviewers' comments.

We look forward to receiving your revised manuscript.

Kind regards,

Anastassia Zabrodskaja, Ph.D.

Academic Editor

PLOS ONE

“This work is part of the projects EDU2017-85909-P and PID2021-123770OB-I00, founded by MCIN/AEI/10.13039/501100011033/FEDER, UE.”

3. We noted in your submission details that a portion of your manuscript may have been presented or published elsewhere. [A very preliminary analysis of the study presented in the manuscript was communicated via poster at the 1st Meeting of the Society for the Advancement of Judgment and Decision Making Studies (Palma, 2016). The poster can be accessed in the following Open Science Framework (OSF) link (not anonymized): https://osf.io/68m2y/?view_only=25d2a1cfbdcf4c349cf10ebebb815104 ] Please clarify whether this [conference proceeding or publication] was peer-reviewed and formally published. If this work was previously peer-reviewed and published, in the cover letter please provide the reason that this work does not constitute dual publication and should be included in the current manuscript.

4. We note that Figure 1 in your submission contain copyrighted images. All PLOS content is published under the Creative Commons Attribution License (CC BY 4.0), which means that the manuscript, images, and Supporting Information files will be freely available online, and any third party is permitted to access, download, copy, distribute, and use these materials in any way, even commercially, with proper attribution. For more information, see our copyright guidelines: http://journals.plos.org/plosone/s/licenses-and-copyright.

Reviewers' comments:

Reviewer's Responses to Questions

**Comments to the Author**

1. Is the manuscript technically sound, and do the data support the conclusions?

Reviewer #1: No

Reviewer #2: Yes

Reviewer #3: Partly

2. Has the statistical analysis been performed appropriately and rigorously? 

Reviewer #1: No

Reviewer #2: Yes

Reviewer #3: Yes

3. Have the authors made all data underlying the findings in their manuscript fully available?

Reviewer #1: Yes

Reviewer #2: Yes

Reviewer #3: No

4. Is the manuscript presented in an intelligible fashion and written in standard English?

Reviewer #1: Yes

Reviewer #2: Yes

Reviewer #3: Yes

5. Review Comments to the Author

Reviewer #1: Thank you for conducting this research and for the analyses you have done. In my point of view, this manuscript is not germane to the journal scope and it can be submitted in a political journal. Your paper quality is fine, and you have met the standards of the academic writing, but it is not suitable to be published here. I do recommend you to find another journal which scope is in harmony with your purpose of study. The next point is that, for your next submission, I ask you to add some graphs or charts in which you show your results to make it more readable.

Reviewer #2: The Foreign-Language (FLe) effect on moral judgments in early bilinguals is evaluated in this study, as well as the potential role of personality traits and emotional load in this type of judgments. The authors conclude that FLe is not evident in early bilinguals, and cruelty is a personality trait than can modulate moral judgements and decisions, regardless of language or dilemma type. They also found significant interactions between some characteristics of dilemmas and personality traits. The manuscript is well written, the design and methodology are appropriate, and the overall discussion is well grounded in the results. The results are interesting for publication. However, there is something that should be discussed in the article before publication. In my opinion the two languages evaluated in this study are very similar (Spanish and Catalan), and they share identical or similar words. In contrast, FLe studies analyze very different languages (English vs. Dutch, or English vs. Chinese, for example). This limitation affecting the results should be critically discussed in the manuscript, and the conclusions from the results should be tone down accordingly. I add here some comments that also might improve the quality of this manuscript.

Abstract

The authors may considerer the possibility to clarify from the abstract if moral dilemmas were displayed on paper or by computer, and it would also be interesting to know here if the response was dichotomous (yes or no) or otherwise.

Introduction

In the Introduction section, the authors should explain the reason to include two similar languages when evaluating the FLe. Have been previously compared these two close related languages in other FLe studies? In my opinion this limitation is critical for understanding the results.

Page 3, lines 53-55: I am not sure if this is the best description of the nature and content of moral dilemmas used in research: “This kind of stimuli describe a situation in which two moral reasons are contraposed: for instance, not to kill (deontological outcome) versus maximizing welfare by any means (utilitarian outcome).”

To understand what exactly means a moral dilemma in moral judgments studies, the authors should improve the provided description of moral dilemmas task, or the meaning and utility of this research tool will be lost.

Page 3, lines 61-63: The authors should clarify that personal dilemmas are usually considered high conflict dilemmas in the moral judgement studies, and impersonal dilemmas are considered low conflict dilemmas. This clarification is in line with the terminology commonly used in this area. This is important when reporting in the results section that impersonal dilemmas (that is, low conflict dilemmas) are more associated with utilitarian responses.

Page 4, lines 83-85: The authors state: “Nevertheless, it seems plausible to say that, for early and proficient bilinguals, L2 should be as emotional as L1; thus, FLe should be absent, because L2 is not a foreign language for simultaneous early bilinguals”

This hypothesis makes sense, but I insist that in this study there is another factor to be considered. L2 might not be a foreign language to the participants of this study as they are early bilinguals, as the authors state, but, most importantly, both languages share words, origins, and phonological and syntactic structures that make them languages not clearly differentiated. This last factor is something that clearly differs from the FLe studies, and should be treated accordingly in this study. In lines 86-87 the authors superficially mention this possibility (“linguistic similarity”), but it should be explicit from the introduction (and detailed in the discussion section).

Page 5, lines 109-110: “when the person who is killed were going to die anyway”; there is a typo here (the person… were…).

Page 5, line 118: “…are thus of particular interest”. Is “are” referring to additional research? In this case, should not be “is thus of…”?

Page 6, lines 141-142: “…thus, personal dilemmas will yield quicker responses than impersonal ones”. I would say that it depends; personal dilemmas being high conflict dilemmas should not be related with faster responses. Maybe the authors mean that personal dilemmas with low conflict will have faster responses. In fact, DT (Reaction Time for decision) should be analyzed differentially in low and high conflict dilemmas, and in personal and impersonal dilemmas. This should be the correct approach to elucidate if DT is affected according to the different dilemma combinations, that is, personal vs. impersonal and high vs. low dilemmas. In case that the authors only use in their study personal dilemmas with low conflict (which on the other hand is not usual), this should be clarify. In most studies, personal dilemmas involve high conflict.

Page 6, lines 143-145: “…we expect no differences between responses to moral dilemmas presented in L2 with respect to L1, as our participants are high proficient early bilinguals…” Again, not only for that reason, but also because the languages evaluated in this study are closely related phonological and syntactically, unlike typical FLe studies (English vs. German, Chinese, etc.). Spanish is considered in this study L2, but participants are actually all Spanish, and it sounds that here there is not a real distinction between L1 and L2. Actually, all of them learned both languages before 3 years of age. How can be distinguished L1 and L2 under this language learning circumstances?

Participants

I am not sure if participants with age of 46 years face the task of this study as students between 18-23 years old would do. Are there statistical comparisons between age sub-groups?

Participants were considered themselves bilingual… Was this confirmed by the authors, apart from the self-reported scales?

Page 8, lines 190-191: “Participants were proficient in both languages, M = 3.67, SD = 0.37, and M = 3.25, SD = 0.57 for L1 and L2 overall proficiency, respectively”. This slightly suggests which one could be L1, but these results are based on self-reported scales. This limitation should be seriously discussed.

Page 9, line 200: “…in accordance with the Declaration of Helsinki”. The last one?

Page 9, lines 201-202: “Authors had no access to information that could identify individual participants during or after data collection”. Therefore, the participation in this study was anonymous, and thus should be reported.

Materials

Page 10 (moral dilemmas task), lines 219-220: “In each of these two categories, four of the eight scenarios were self-beneficial (SB) and four were other-beneficial dilemmas (OB)”. Does this mean low and high conflict dilemmas? What were low and high conflict dilemmas should be clarify.

Page 10, lines 228-229: I was not able to access to the files by the link https://osf.io/4vry5/?view_only=597aa6dab16c43dd9d32b6036691958b

Are they really available?

Procedure

Page 11, line 236: This is the first time that readers can know if dilemmas were displayed on paper or otherwise. In my opinion, this information should be briefly mentioned from the beginning (the abstract), as previous studies have generally used a dilemma presentation in paper format.

Page 11, lines 239-241: “A block of 10 dilemmas in Catalan and a block of 10 dilemmas in Spanish, counterbalancing the order of language blocks across participants, were presented by E-Prime 2.0 [44] to all participants”. Do you mean counterbalancing the order of language blocks and in a randomized way across participants? Otherwise, each counterbalancing result (Spanish-Catalan, and Catalan-Spanish) is supposed to be presented continuously in half of participants (first S-C and then C-S, or the other way).

Page 11, lines 249-250: “responding by pressing I key (meaning "yes") or O key (meaning "no")”. Can the authors explain why these keys were used? Responses were made with a particular hand (right or left) or the participants responded indistinctly with either hand. This is important, among other things, for RT (DT).

Page 12, line 261: again, not files can be downloaded from the link provided by the authors: https://osf.io/4vry5/?view_only=597aa6dab16c43dd9d32b6036691958b

Statistical analyses

Page 12, lines 277-278: “Responses to moral dilemmas with DT < 1000 ms (2.4% of trials) were excluded from the analyses”. That means that DTs of, for example, 1.1 sc were included. Is this time enough to read the question (two lines according to Fig. 1) and take the decision?

Page 16, Table 3 legend: “a: One participant answered that he/she was slightly more proficient in L2. It seems a mistake but some early bilinguals in Mallorca consider that they are equally or more proficient in L2 because their true mother tongue differed from the Catalan normative language”. Therefore, it is not clear even to the participants which one is L1 and L2, and the interpretation of the results of this study should consider this limitation.

Discussion

Please, consider the limitation regarding the similarity of both languages when discussing the results related to the FLe.

Page 28, lines 599-601: “Future studies could address these limitations by using larger and more diverse samples or investigating other factors that could affect moral decision-making, such as cultural background or education level”. The authors may want to discuss the cultural and educational effect on moral decisions in line with the results reported in the following study:

https://www.tandfonline.com/doi/full/10.1080/00049530.2021.1882276

Moreover, a gradation in the responses (which were not dichotomous) can be found in that study, which has not been previously evaluated in the moral dilemma literature. The authors may also want to discuss this procedural difference (a graduation in the responses) in regards to their results.

A last minor thing. The short title is “MEANNESS AND UTILITARIAN JUDGMENTS IN EARLY BILINGUALS”, but I think that utilitarian are the responses (or solutions to dilemmas), not the judgements themselves.

Reviewer #3: This study investigates a relevant topic (decision-making and individual differences). However, it tries to combine so many variables that it is difficult to maintain the focus and overall coherence and to provide an in-depth interpretation of the results. The novelty of the study is not clear; it appears that replicating previous findings is the main aim of the study. For example, we already know that proficiency level influences decision-making. In addition, we know that for many L1 speakers of Catalan, Spanish turns out to be the language that they mostly use or are exposed to much more often than their L1, which limits the conclusions that can be derived from this study. In what follows, I provide more detailed comments.

INTRODUCTION

- In the introduction, the authors argue that “language can make a difference in our moral decisions … because language is unavoidably related with emotion”. That´s correct. But it is surprising the fact that the only study that the authors cite is Brouwer (2021). I believe that other studies, especially those that used mediation analyses to examine the mediating role of emotions in the link between language and moral decision-making should be acknowledged; see, for example,

• Geipel, J., Hadjichristidis, C., & Surian, L. (2015a). How foreign language shapes moral judgment. Journal of Experimental Social Psychology, 59, 8–17. https://doi.org/10.1016/j.jesp.2015.02.001

• Geipel, J., Hadjichristidis, C., & Surian, L. (2015b). The foreign language effect on moral judgment: The role of emotions and norms. PloS One, 10(7), Article e0131529. https://doi.org/10.1371/journal.pone.0131529

• Kyriakou, A., Foucart, A., & Mavrou, I. (2022). Moral judgments in a foreign language: Expressing emotions and justifying decisions. International Journal of Bilingualism. https://doi.org/10.1177/13670069221134193

- The same applies to “In this vein, FLe studies with moral dilemmas [3,10] have suggested that second language provides to bilingual people a greater emotional distance than their first, native language (L1), leading to more utilitarian decisions using L2 than using L1”. Please consider adding more relevant studies.

- “It has been said that bilinguals are more emotional before a moral dilemma when…” – Why before and not after?

- “There is certain controversy about their presence [of the FLe] in early, high proficient, bilinguals” – That´s correct. The authors may want to include more studies that investigated the role of proficiency level, such as:

• Corey, J. D., Hayakawa, S., Foucart, A., Aparici, M., Botella, J., Costa, A., & Keysar, B. (2017). Our moral choices are foreign to us. Journal of Experimental Psychology: Learning, Memory, and Cognition, 43(7), 1109–1128. https://doi.org/10.1037/xlm0000356

• Kirova, A., Tang, Y., & Conway, P. (2023). Are people really less moral in their foreign language? Proficiency and comprehension matter for the moral foreign language effect in Russian speakers. PloS One, 18(7), Article e0287789.

Based on this research, the authors must justify the contribution and novelty of their study in a more explicit and straightforward way.

GOALS OF THE STUDY

- The first goal of the study is to replicate previous findings. Please explain why this replication is important and necessary and how this replication adds to what we already know about the topic.

- The second goal is to “explore how psychopathic traits and variables of the dilemmas would separately and/or jointly affect moral judgment behavior”. First, it is not clear how this goal is linked to the first one. Second, you could name the “variables of the dilemmas”. Third, please explain better what you mean by “separately and/or jointly”. Fourth, personal force, benefit recipient and avoidability have not been defined until line 130. Fifth, difficulty, verisimilitude and vividness are introduced in lines 131–132 without previous explanations of why these variables have been chosen or what the theoretical framework behind this choice is. Could you please link better this objective to your theoretical framework? Does objective 2 also aims to replicate previous findings?

METHOD

Participants

- How did you make sure that your participants were balanced bilinguals? Just by asking them?

- “Gender” would be preferable to “sex”.

- You consider that Catalan was participants’ L1. However, is Catalan the language that the participants were mainly exposed to and communicated in? Descriptive statistics were slightly higher for L2, which makes one think that Spanish might have been the dominant language.

- 75% of the participants were females. We know that gender differences may exist when it comes to moral decision-making. How did you control for gender differences?

Materials

- Please define “boldness”, “meanness” and “disinhibition”.

- Please upload the dilemmas and scales on OSF. Please correct me if I am wrong, but in the link you provide only a .sav file is included, and therefore it is impossible to assess the dilemmas you have used in your study.

- How many situations/dilemmas did you finally use? 16 or 22?

RESULTS

- How do you explain the fact that personal and impersonal dilemmas obtained almost the same descriptive values for valence and arousal?

- The values for arousal were quite similar. Why did you include arousal as a covariate? And why did some models include arousal while others both arousal and valence?

- Why did you include differences in proficiency between L1 and L2 as a covariate? Usually, L1 Catalan people are highly proficient in their L2 Spanish, which they might be using even more often than Catalan (see my previous comments).

- “We decided to explore a little bit more the data” -- What was the rationale behind this decision? Doing additional analyses without a reason seems unjustified, particularly because you did not preregister your study.

DISCUSSION

- Could you please elaborate a bit more on the following interpretation “Moreover, contrary to Wong and Ng [18] study, our results show no arousal interaction with personal force, supporting the Dual Process Model (DPM) by Greene and colleagues”?

- You say that considering arousal is crucial, but you also add that arousal did not modulate the effect of other predictors, nor did it predict the amount of time consumed, so taking into account arousal is crucial “when”?

- Most part of the discussion repeats the results and includes comparisons with previous findings, i.e. interpretations of your results are missing. For example, how do you explain the fact that your participants tended to respond more utilitarianly when they perceived more control and when the decision was perceived as more difficult? How do you explain the fact that dilemmas perceived as difficult and vivid led to slower DT? How do you interpret the interaction of boldness and dominance, or the interaction of boldness, meanness and disinhibition?, etc. As I mentioned previously, you have tried to combine and simultaneously analyse many different variables, and as a result elaborating a coherent and in-depth discussion of the main findings appears to be very challenging.

- I would also like to add that what you have found in your study is similar to what has been found in previous studies with unbalanced bilinguals and realistic moral dilemmas, i.e. the lack of the moral foreign language effect. In other words, the moral language effect is not the "norm", unless unrealistic moral dilemmas (e.g. the footbridge dilemma) are used, and between this is something that you would like to elaborate on in your paper; see, for example:

• Białek, M., Paruzel-Czachura, M., & Gawronski, B. (2019). Foreign language effects on moral dilemma judgments: An analysis using the CNI model. Journal of Experimental Social Psychology, 85, Article 103855.

• Kyriakou, A., & Mavrou, I. (2023). ¿Eres muy emocional? I don’t think so. How does language determine our emotional responses to everyday moral dilemmas? In A. Blanco Canales & S. Martín Leralta (Eds.), Emotion and identity in second language learning (pp. 297–321). Peter Lang.

• Muda, R., Niszczota, P., Białek, M., & Conway, P. (2018). Reading dilemmas in a foreign language reduces both deontological and utilitarian response tendencies. Journal of Experimental Psychology: Learning, Memory, and Cognition, 44(2), 321–326.

LANGUAGE

The manuscript would benefit from a proofreading (e.g. “considered themselves bilingual*”, “were neither* significant predictors”, “let us to* find”, “both a high experience of unpleasantness and a high experience of pleasantness leads* to ..”, etc.).

I am aware that I have raised many concerns, but I am convinced that if the authors clearly explain the contribution of their study and justify some methdodological decisions and interpretations, this paper could be a nice contribution to the literature on the moral foreign language effect.

6. PLOS authors have the option to publish the peer review history of their article (what does this mean?). If published, this will include your full peer review and any attached files.

Reviewer #1: **Yes: **Dr. Farzaneh Shakki

Reviewer #2: **Yes: **Andres Molero-Chamizo

Reviewer #3: No

---

## [Author Response · Author response to Decision Letter 0]

27 Sep 2023

We sincerely appreciate the time and effort you have invested in reviewing our manuscript. We have carefully addressed each of your comments and suggestions in the rebuttal letter (Response to reviewers.docx), making necessary revisions to the manuscript to ensure its accuracy and comprehensibility. We are confident that the revised version now better aligns with the standards and expectations of the journal.

Thank you again for your time and consideration.

---

## [Decision Letter · Decision Letter 1]

2 Oct 2023

PONE-D-23-10940R1Meanness trumps language: Lack of foreign language effect in early bilinguals' moral choicesPLOS ONE

Dear Dr. Flexas,

Thank you for submitting your manuscript to PLOS ONE. After careful consideration, we feel that it has merit but does not fully meet PLOS ONE’s publication criteria as it currently stands. Therefore, we invite you to submit a revised version of the manuscript that addresses the points raised during the review process.

ACADEMIC EDITOR: Please implement these minor changes.

We look forward to receiving your revised manuscript.

Kind regards,

Anastassia Zabrodskaja, Ph.D.

Academic Editor

PLOS ONE

Journal Requirements:

Reviewers' comments:

Reviewer's Responses to Questions

**Comments to the Author**

1. If the authors have adequately addressed your comments raised in a previous round of review and you feel that this manuscript is now acceptable for publication, you may indicate that here to bypass the “Comments to the Author” section, enter your conflict of interest statement in the “Confidential to Editor” section, and submit your "Accept" recommendation.

Reviewer #1: All comments have been addressed

Reviewer #2: All comments have been addressed

2. Is the manuscript technically sound, and do the data support the conclusions?

Reviewer #1: Yes

Reviewer #2: Partly

3. Has the statistical analysis been performed appropriately and rigorously? 

Reviewer #1: Yes

Reviewer #2: Yes

4. Have the authors made all data underlying the findings in their manuscript fully available?

Reviewer #1: Yes

Reviewer #2: Yes

5. Is the manuscript presented in an intelligible fashion and written in standard English?

Reviewer #1: Yes

Reviewer #2: Yes

6. Review Comments to the Author

Reviewer #1: Hi. Thank you for the revised files. The comments are done carefully. I am happy with the revised file.

Reviewer #2: The authors have responded to each of my comments, and I highly appreciate that. However, I still have a concern regarding the authors's perception of the two languages evaluated in their study. They state that Catalan (that they considered L1) is a "native language" for the participants of the study, and Spanish is an "early acquired yet distinct (not native) language". And, if I understood properly, this perception was based on self-reports, and was supposedly "confirmed" by language proficiency scores. I believe that authors should tone down a bit more these statements through the article, and do not assume that slightly higher proficiency scores are a reliable proof or evidence of the distinction between "native" and "not native" languages. This has also been criticized by reviewer 3, and I think that the authors should reflect on this limitation regarding which language is "native" or not. Obviously, Spanish is the native language in Spain.

A last minor question I have to the authors is if the 46 years old participant was also university students, as the authors state that all participants of the study were university students. If not, and considering the huge age deviation with respect to the rest of the participants, would not be more precise, statistically and echologically, only include in the analysis the young sample? If the authors keep in the analysis, for any reason, the data of the 46 years old participant, this limitation should also be discussed in the Discussion section.

7. PLOS authors have the option to publish the peer review history of their article (what does this mean?). If published, this will include your full peer review and any attached files.

Reviewer #1: **Yes: **Dr. Farzaneh Shakki

Reviewer #2: **Yes: **Andrés Molero-Chamizo

---

## [Author Response · Author response to Decision Letter 1]

11 Oct 2023

As requested, we include a rebuttal letter with responses to reviewers and a marked-up copy of our manuscript with Microsoft Word tracked changes.

---

## [Decision Letter · Decision Letter 2]

3 Nov 2023

Meanness trumps language: Lack of foreign language effect in early bilinguals' moral choices

PONE-D-23-10940R2

Dear Dr. Flexas,

We’re pleased to inform you that your manuscript has been judged scientifically suitable for publication and will be formally accepted for publication once it meets all outstanding technical requirements.

Kind regards,

Anastassia Zabrodskaja, Ph.D.

Academic Editor

PLOS ONE

Additional Editor Comments (optional):

Reviewers' comments:

Reviewer's Responses to Questions

**Comments to the Author**

1. If the authors have adequately addressed your comments raised in a previous round of review and you feel that this manuscript is now acceptable for publication, you may indicate that here to bypass the “Comments to the Author” section, enter your conflict of interest statement in the “Confidential to Editor” section, and submit your "Accept" recommendation.

Reviewer #2: All comments have been addressed

2. Is the manuscript technically sound, and do the data support the conclusions?

Reviewer #2: Partly

3. Has the statistical analysis been performed appropriately and rigorously? 

Reviewer #2: Yes

4. Have the authors made all data underlying the findings in their manuscript fully available?

Reviewer #2: Yes

5. Is the manuscript presented in an intelligible fashion and written in standard English?

Reviewer #2: Yes

6. Review Comments to the Author

Reviewer #2: The authors have addressed all my comments, and I believe this manuscript metts in its present form the scientific requirements to be published

7. PLOS authors have the option to publish the peer review history of their article (what does this mean?). If published, this will include your full peer review and any attached files.

Reviewer #2: **Yes: **Andrés Molero-Chamizo

---

## [Editor Report · Acceptance letter]

17 Nov 2023

PONE-D-23-10940R2 

Meanness trumps language: Lack of foreign language effect in early bilinguals' moral choices 

Dear Dr. Flexas:

I'm pleased to inform you that your manuscript has been deemed suitable for publication in PLOS ONE. Congratulations! Your manuscript is now with our production department. 

Kind regards, 

on behalf of

Professor Anastassia Zabrodskaja 

Academic Editor

PLOS ONE